

# Complexity curve: a graphical measure of data complexity and classifier performance

Julian Zubek[1,2] and Dariusz M. Plewczynski[1]

[1] Centre of New Technologies, University of Warsaw, Warsaw, Poland
[2] Institute of Computer Science, Polish Academy of Sciences, Warsaw, Poland

## ABSTRACT

We describe a method for assessing data set complexity based on the estimation of the underlining probability distribution and Hellinger distance. In contrast to some popular complexity measures, it is not focused on the shape of a decision boundary in a classification task but on the amount of available data with respect to the attribute structure. Complexity is expressed in terms of graphical plot, which we call complexity curve. It demonstrates the relative increase of available information with the growth of sample size. We perform theoretical and experimental examination of properties of the introduced complexity measure and show its relation to the variance component of classification error. We then compare it with popular data complexity measures on 81 diverse data sets and show that it can contribute to explaining performance of specific classifiers on these sets. We also apply our methodology to a panel of simple benchmark data sets, demonstrating how it can be used in practice to gain insights into data characteristics. Moreover, we show that the complexity curve is an effective tool for reducing the size of the training set (data pruning), allowing to significantly speed up the learning process without compromising classification accuracy. The associated code is available to download at: https://github.com/zubekj/complexity_curve (open source Python implementation).

# INTRODUCTION

It is common knowledge in machine learning community that the difficulty of classification problems varies greatly. Sometimes it is enough to use a simple out-of-the-box classifier to get a very good result and sometimes careful preprocessing and model selection are needed to get any non-trivial result at all. The difficulty of a classification task clearly stems from certain properties of the data set, yet we still have problems with defining those properties in general.

Bias–variance decomposition (*Domingos, 2000*) demonstrates that the error of a predictor can be attributed to three sources: bias, coming from the inability of an algorithm to build an adequate model for the relationship present in data; variance, coming from the inability to estimate correct model parameters from an imperfect data sample; and the

Corresponding author
Dariusz M. Plewczynski,
dariuszplewczynski@gmail.com

irreducible error component commonly called noise. Following this line of reasoning, the difficulty of a classification problem may come partly from the complexity of the relation between dependent variable and explanatory variables, partly from the scarcity of information in the training sample, and partly from class ambiguity (due to noise in the target variable or an overlap between classes). This is identical to sources of classification difficulty identified by *Ho & Basu (2002)*, who labelled the three components: 'complex decision boundary,' 'small sample size and dimensionality induced sparsity' and 'ambiguous classes.'

In this article, we introduce a new measure of data complexity targeted at sample sparsity, which is mostly associated with variance error component. We aim to measure information saturation of a data set without making any assumptions on the form of the relation between dependent variable and the rest of variables, so explicitly disregarding shape of the decision boundary and classes ambiguity (e.g., caused by noise on the target variable). Our complexity measure takes into account the number of samples, the number of attributes, and the internal structure of attributes under a simplifying assumption of attribute independence. The key idea is to check how well a data set can be approximated by its subsets. If the probability distribution induced by a small data sample is very similar to the probability distribution induced by the whole data set, we say that the set is saturated with information and presents an opportunity to learn the relationship between variables without promoting the variance. To operationalise this notion, we introduce two kinds of plots:

- Complexity curve—a plot presenting how well subsets of growing size approximate distribution of attribute values. It is a basic method applicable to clustering, regression and classification problems.
- Conditional complexity curve—a plot presenting how well subsets of growing size approximate conditional distribution of attribute values given class. It is applicable to classification problems and more robust against class imbalance or differences in attributes structure between classes.

Since the proposed measure characterise the data sample itself without making any assumptions as to how that sample will be used, it should be applicable to all kinds of problems involving reasoning from data. In this work, we focus on classification tasks since this is the context in which data complexity measures were previously applied. We compare the area under the complexity curve with popular data complexity measures and show how it complements the existing metrics. We also demonstrate that it is useful for explaining classifier performance by showing that the area under the complexity curve is correlated with the area under the receiver operating characteristic (AUC ROC) for popular classifiers tested on 81 benchmark data sets.

We propose an immediate application of the developed method connected with the fundamental question: how large data sample is needed to build a successful predictor? We pursue this topic by proposing a data pruning strategy based on complexity curve and evaluating it on large data sets. We show that it can be considered as an alternative to progressive sampling strategies (*Provost, Jensen & Oates, 1999*).

## RELATED LITERATURE

The problem of measuring data complexity in the context of machine learning is broadly discussed. Our beliefs are similar to those of *Ho (2008)*, who stated the need for including data complexity analysis in algorithm comparison procedures. Similar needs are also discussed in fields outside machine learning; for example, in combinatorial optimisation (*Smith-Miles & Lopes, 2012*).

The general idea is to select a sufficiently diverse set of problems to demonstrate both strengths and weaknesses of the analysed algorithms. The importance of this step was stressed by *Macià et al. (2013)*, who demonstrated how algorithm comparison may be biased by benchmark data set selection, and showed how the choice may be guided by complexity measures. Characterising problem space with some metrics makes it possible to estimate regions in which certain algorithms perform well (*Luengo & Herrera, 2013*), and this opens up possibilities of meta-learning (*Smith-Miles et al., 2014*).

In this context, complexity measures are used not only as predictors of classifier performance but, more importantly, as diversity measures capturing various properties of the data sets. It is useful when the measures themselves are diverse and focus on different aspects of the data to give as complete characterisation of the problem space as possible. In the later part of the article we demonstrate that complexity curve fits well into the landscape of currently used measures, offering new insights into data characteristics.

A set of practical measures of data complexity with regard to classification was introduced by *Ho & Basu (2002)*, and later extended by *Ho, Basu & Law (2006)* and *Orriols-Puig, Macià & Ho (2010)*. It is routinely used in tasks involving classifier evaluation (*Macià et al., 2013*; *Luengo & Herrera, 2013*) and meta-learning (*Diez-Pastor et al., 2015*; *Mantovani et al., 2015*). Some of these measures are based on the overlap of values of specific attributes; examples include Fisher's discriminant ratio, volume of overlap region, attribute efficiency etc. The others focus directly on class separability; this group includes measures such as the fraction of points on the decision boundary, linear separability, the ratio of intra/inter class distance. In contrast to our method, such measures focus on specific properties of the classification problem, measuring shape of the decision boundary and the amount class overlap. Topological measures concerned with data sparsity, such as ratio of attributes to observations, attempt to capture similar properties as our complexity curve.

*Li & Abu-Mostafa (2006)* defined data set complexity in the context of classification using the general concept of Kolmogorov complexity. They proposed a way to measure data set complexity using the number of support vectors in support vector machine (SVM) classifier. They analysed the problems of data decomposition and data pruning using above methodology. A graphical representation of the data set complexity called the complexity-error plot was also introduced. The main problem with their approach is the selection of very specific and complex machine learning algorithms, which may render the results in less universal way, and which are prone to biases specific to SVMs. This make their method unsuitable for diverse machine learning algorithms comparison.

Another approach to data complexity is to analyse it on the level of individual instances. This kind of analysis is performed by *Smith, Martinez & Giraud-Carrier (2013)*, who

attempted to identify which instances are misclassified by various classification algorithm. They devised local complexity measures calculated with respect to single instances and later tried to correlate average instance hardness with global data complexity measures of *Ho & Basu (2002)*. They discovered that it is mostly correlated with class overlap. This makes our work complementary, since in our complexity measure we deliberately ignore class overlap and individual instance composition to isolate another source of difficulty, namely data scarcity.

*Yin et al. (2013)* proposed a method of feature selection based on Hellinger distance (a measure of similarity between probability distributions). The idea was to choose features, which conditional distributions (depending on the class) have minimal affinity. In the context of our framework, this could be interpreted as measuring data complexity for single features. The authors demonstrated experimentally that, for the high-dimensional imbalanced data sets, their method is superior to popular feature selection methods using Fisher criterion, or mutual information.

## DEFINITIONS

In the following sections, we define formally all measures used throughout the paper. Basic intuitions, assumptions, and implementation choices are discussed. Finally, algorithms for calculating complexity curve, conditional complexity curve, and generalisation curve are given.

### Measuring data complexity with samples

In a typical machine learning scenario, we want to use information contained in a collected data sample to solve a more general problem which our data describe. Problem complexity can be naturally measured by the size of a sample needed to describe the problem accurately. We call the problem complex, if we need to collect a lot of data in order to get any results. On the other hand, if a small amount of data suffices, we say the problem has low complexity.

How to determine if a data sample describes the problem accurately? Any problem can be described with a multivariate probability distribution $P$ of a random vector $X$. From $P$ we sample our finite data sample $D$. Now, we can use $D$ to build the estimated probability distribution of $X$–$P_D$. $P_D$ is the approximation of $P$. If $P$ and $P_D$ are identical, we know that data sample $D$ describes the problem perfectly and collecting more observations would not give us any new information. Analogously, if $P_D$ is very different from $P$ we can be almost certain that the sample is too small.

To measure similarity between probability distributions we use Hellinger distance. For two continuous distributions $P$ and $P_D$ with probability density functions $p$ and $p_D$ it is defined as:

$$H^2(P, P_D) = \frac{1}{2} \int \left( \sqrt{p(x)} - \sqrt{p_D(x)} \right)^2 dx.$$

The minimum possible distance 0 is achieved when the distributions are identical, the maximum 1 is achieved when any event with non-zero probability in $P$ has probability 0 in $P_D$ and vice versa. Simplicity and naturally defined 0–1 range make Hellinger distance a good measure for capturing sample information content.

In most cases, we do not know the underlining probability distribution $P$ representing the problem and all we have is a data sample $D$, but we can still use the described complexity measure. Let us picture our data $D$ as the true source of knowledge about the problem and the estimated probability distribution $P_D$ as the reference distribution. Any subset $S \subset D$ can be treated as a data sample and a probability distribution $P_S$ estimated from it will be an approximation of $P_D$. By calculating $H^2(P_D, P_S)$ we can assess how well a given subset represent the whole available data, i.e., determine its information content.

Obtaining a meaningful estimation of a probability distribution from a data sample poses difficulties in practice. The probability distribution we are interested in is the joint probability on all attributes. In that context, most of the realistic data sets should be regarded as extremely sparse, and naïve probability estimation using frequencies of occurring values would result in mostly flat distribution. This can be called the curse of dimensionality. Against this problem, we apply a naïve assumption that all attributes are independent. This may seem like a radical simplification but, as we will demonstrate later, it yields good results in practice and constitute a reasonable baseline for common machine learning techniques. Under the independence assumption we can calculate the joint probability density function $f$ from the marginal density functions $f_1, \ldots, f_n$:

$$f(x) = f_1(x_1)f_2(x_2)\ldots f_n(x_n).$$

We will now show the derived formula for Hellinger distance under the independence assumption. Observe that the Hellinger distance for continuous variables can be expressed in another form:

$$\frac{1}{2}\int \left(\sqrt{f(x)} - \sqrt{g(x)}\right)^2 dx = \frac{1}{2}\int \left(f(x) - 2\sqrt{f(x)g(x)} + g(x)\right) dx$$
$$= \frac{1}{2}\int f(x)\, dx - \int \sqrt{f(x)g(x)}\, dx + \frac{1}{2}\int g(x)\, dx$$
$$= 1 - \int \sqrt{f(x)g(x)}\, dx.$$

In the last step we used the fact the that the integral of a probability density over its domain must equal one.

We will consider two multivariate distributions $F$ and $G$ with density functions:

$$f(x_1, \ldots, x_n) = f_1(x_1)\ldots f_n(x_n)$$
$$g(x_1, \ldots, x_n) = g_1(x_1)\ldots g_n(x_n).$$

The last formula for Hellinger distance will now expand:

$$1 - \int \cdots \int \sqrt{f(x_1, \ldots, x_n)g(x_1, \ldots, x_n)}dx_1\ldots dx_n$$
$$= 1 - \int \cdots \int \sqrt{f_1(x_1)\ldots f_n(x_n)g_1(x_1)\ldots g_n(x_n)}dx_1\ldots dx_n$$
$$= 1 - \int \sqrt{f_1(x_1)g_1(x_1)}\, dx_1 \ldots \int \sqrt{f_n(x_n)g_n(x_n)}\, dx_n.$$

In this form, variables are separated and parts of the formula can be calculated separately.

## Practical considerations

Calculating the introduced measure of similarity between data set in practice poses some difficulties. First, in the derived formula direct multiplication of probabilities occurs, which leads to problems with numerical stability. We increased the stability by switching to the following formula:

$$1 - \int \sqrt{f_1(x_1)g_1(x_1)}\, dx_1 \ldots \int \sqrt{f_n(x_n)g_n(x_n)}\, dx_n$$
$$= 1 - \left(1 - \frac{1}{2}\int \left(\sqrt{f_1(x_1)} - \sqrt{g_1(x_1)}\right)^2 dx_1\right) \ldots \left(1 - \frac{1}{2}\int \left(\sqrt{f_n(x_n)} - \sqrt{g_n(x_n)}\right)^2 dx_2\right)$$
$$= 1 - \left(1 - H^2(F_1, G_1)\right) \ldots \left(1 - H^2(F_n, G_n)\right).$$

For continuous variables probability density function is routinely done with kernel density estimation (KDE)—a classic technique for estimating the shape continuous probability density function from a finite data sample (*Scott, 1992*). For a sample $(x_1, x_2, \ldots, x_n)$ estimated density function has a form:

$$\hat{f}_h(x) = \frac{1}{nh}\sum_{i=1}^{n} K\left(\frac{x - x_i}{h}\right)$$

where $K$ is the kernel function and $h$ is a smoothing parameter –bandwidth. In our experiments we used Gaussian function as the kernel. This is a popular choice, which often yields good results in practice. The bandwidth was set according to the modified Scott's rule (*Scott, 1992*):

$$h = \frac{1}{2}n^{-\frac{1}{d+4}},$$

where $n$ is the number of samples and $d$ number of dimensions.

In many cases, the independence assumption can be supported by preprocessing input data in a certain way. A very common technique, which can be applied in this situation is the whitening transform. It transforms any set of random variables into a set of uncorrelated random variables. For a random vector $X$ with a covariance matrix $\Sigma$ a new uncorrelated vector $Y$ can be calculated as follows:

$$\Sigma = PDP^{-1}$$
$$W = PD^{-\frac{1}{2}}P^{-1}$$
$$Y = XW$$

where $D$ is diagonal matrix containing eigenvalues and $P$ is matrix of right eigenvectors of $\Sigma$. Naturally, lack of correlation does not imply independence but it nevertheless reduces the error introduced by our independence assumption. Furthermore, it blurs the difference between categorical variables and continuous variables putting them on an equal footing. In all further experiments, we use whitening transform preprocessing and then treat all variables as continuous.

A more sophisticated method is a signal processing technique known as Independent Component Analysis (ICA) (*Hyvärinen & Oja, 2000*). It assumes that all components of an observed multivariate signal are mixtures of some independent source signals and that the distribution of the values in each source signal is non-Gaussian. Under these assumption, the algorithm attempts to recreate the source signals by splitting the observed signal into the components as independent as possible. Even if the assumptions are not met, ICA technique can reduce the impact of attributes interdependencies. Because of its computational complexity, we used it as an optional step in our experiments.

## Machine learning task difficulty

Our data complexity measure can be used for any type of problem described through a multivariate data sample. It is applicable to regression, classification and clustering tasks. The relation between the defined data complexity and the difficulty of a specific machine learning task has to be investigated. We will focus on the supervised learning case. Classification error will be measured as mean 0–1 error (accuracy). Data complexity will be measured as mean Hellinger distance between the real and the estimated probability distributions of attributes conditioned on the target variable:

$$\frac{1}{m}\sum_{i=1}^{m}H^2(P(X|Y=y_i),P_D(X|Y=y_i))$$

where $X$—vector of attributes, $Y$—target variable, $y_1,y_2,\ldots y_m$—values taken by $Y$.

It has been shown that error of an arbitrary classification or regression model can be decomposed into three parts:

$$\text{Error} = \text{Bias} + \text{Variance} + \text{Noise}.$$

*Domingos (2000)* proposed an universal scheme of decomposition, which can be adapted for different loss functions. For a classification problem and 0–1 loss $L$ expected error on sample $x$ for which the true label is $t$, and the predicted label given a training set $D$ is $y$ can be expressed as:

$$E_{D,t}[\mathbb{1}(t \neq y)]$$
$$= \mathbb{1}(E_t[t] \neq E_D[y]) + c_2 E_D[\mathbb{1}(y \neq E_D[y])] + c_1 E_t[\mathbb{1}(t \neq E_t[t])]$$
$$= B(x) + c_2 V(x) + c_1 N(x)$$

where $B$—bias, $V$—variance, $N$—noise. Coefficients $c_1$ and $c_2$ are added to make the decomposition consistent for different loss functions. In this case, they are equal to:

$$c_1 = P_D(y = E_t[t]) - P_D(y \neq E_t[t])P_t(y = t \mid E_t[t] \neq t)$$
$$c_2 = \begin{cases} 1 & \text{if } E_t[t] = E_D[y] \\ -P_D(y = E_t[t] \mid y \neq E_D[y]) & \text{otherwise.} \end{cases}$$

Bias comes from an inability of the applied model to represent the true relation present in data, variance comes from an inability to estimate the optimal model parameters from the

data sample, the noise is inherent to the solved task and irreducible. Since our complexity measure is model agnostic, it clearly does not include bias component. As it does not take into account the dependent variable, it cannot measure noise either. All that is left to investigate is the relation between our complexity measure and variance component of the classification error.

The variance error component is connected with overfitting, when the model fixates over specific properties of a data sample and looses generalisation capabilities over the whole problem domain. If the training sample represented the problem perfectly and the model was fitted with perfect optimisation procedure, variance would be reduced to zero. The less representative the training sample is for the whole problem domain, the larger the chance for variance error.

This intuition can be supported by comparing our complexity measure with the error of the Bayes classifier. We will show that they are closely related. Let $Y$ be the target variable taking on values $v_1, v_2, \ldots, v_m$, $f_i(x)$ an estimation of $P(X = x | Y = v_i)$ from a finite sample $D$, and $g(y)$ an estimation of $P(Y = y)$. In such a setting, 0–1 loss of the Bayes classifier on a sample $x$ with the true label $t$ is:

$$\mathbb{1}(t \neq y) = \mathbb{1}\left(t \neq \mathrm{argmax}_i\left(g(v_i)f_i(x)\right)\right).$$

Let assume that $t = v_j$. Observe that:

$$v_j = \mathrm{argmax}_i\left(g(v_i)f_i(x)\right) \Leftrightarrow \forall_i g(v_j)f_j(x) - g(v_i)f_i(x) \geq 0$$

which for the case of equally frequent classes reduces to:

$$\forall_i f_j(x) - f_i(x) \geq 0.$$

We can simultaneously add and subtract term $P(X = x \mid Y = v_j) - P(X = x \mid Y = v_i)$ to obtain:

$$\forall_i \left(f_j(x) - P(X = x \mid Y = v_j)\right) + \left(P(X = x \mid Y = v_i) - f_i(x)\right)$$
$$+ \left(P(X = x \mid Y = v_j) - P(X = x \mid Y = v_i)\right) \geq 0.$$

We know that $P(X = x | Y = v_j) - P(X = x | Y = v_i) \geq 0$, so as long as estimations $f_i(x)$, $f_j(x)$ do not deviate too much from real distributions the inequality is satisfied. It will not be satisfied (i.e., an error will take place) only if the estimations deviate from the real distributions in a certain way (i.e., $f_j(x) < P(X = x | Y = v_j)$ and $f_i(x) > P(X = x | Y = v_i)$) and the sum of these deviations is greater than $P(X = x | Y = v_j) - P(X = x | Y = v_i)$. The Hellinger distance between $f_i(x)$ and $P(X = x | Y = v_i)$ measures the deviation. This shows that by minimising Hellinger distance we are also minimising error of the Bayes classifier. The converse may not be true: not all deviations of probability estimates result in classification error.

In the introduced complexity measure, we assumed independency of all attributes, which is analogous to the assumption of naïve Bayes. Small Hellinger distance between

class-conditioned attribute distributions induced by sets $A$ and $B$ means that naïve Bayes trained on set $A$ and tested on set $B$ will have only very slight variance error component. Of course, if the independence assumption is broken bias error component may still be substantial.

## Complexity curve

Complexity curve is a graphical representation of a data set complexity. It is a plot presenting the expected Hellinger distance between a subset and the whole set versus subset size:

$$CC(n) = E[H^2(P, Q_n)]$$

where $P$ is the empirical probability distribution estimated from the whole set and $Q_n$ is the probability distribution estimated from a random subset of size $n \leq |D|$. Let us observe that $CC(|D|) = 0$ because $P = Q_{|D|}$. $Q_0$ is undefined, but for the sake of convenience we assume $CC(0) = 1$.

---

**Algorithm 1** Procedure for calculating complexity curve.

$D$ – original data set, $K$ – number of random subsets of the specified size.

1. Transform $D$ with whitening transform and/or ICA to obtain $D_I$.
2. Estimate probability distribution for each attribute of $D_I$ and calculate joint probability distribution – $P$.
3. For $i$ in $1\dots|D_I|$ (with an optional step size $d$):
   (a) For $j$ in $1\dots K$:
      i. Draw subset $S_i^j \subseteq D_I$ such that $|S_i^j| = i$.
      ii. Estimate probability distribution for each attribute of $S_i^j$ and calculate joint probability distribution – $Q_i^j$.
      iii. Calculate Hellinger distance: $l_i^j = H^2(P, Q_i^j)$.
   (b) Calculate mean $m_i$ and standard error $s_i$:

$$m_i = \frac{1}{K}\sum_{j=1}^{K} l_i^j \qquad s_i = \sqrt{\frac{1}{K}\sum_{j=1}^{K}\left(m_i - l_i^j\right)^2}$$

Complexity curve is a plot of $m_i \pm s_i$ vs $i$.

---

To estimate complexity curve in practice, for each subset size $K$ random subsets are drawn and the mean value of Hellinger distance, along with standard error, is marked on the plot. The Algorithm 1 presents the exact procedure. Parameters $K$ (the number of samples of a specified size) and $d$ (sampling step size) control the trade-off between the precision of the calculated curve and the computation time. In all experiments, unless stated otherwise, we used values $K = 20$, $d = \frac{|D|}{60}$. Regular shapes of the obtained curves did not suggest the need for using larger values.

Figure 1 presents a sample complexity curve (solid lines). It demonstrates how by drawing larger subsets of the data we get better approximations of the original distribution, as indicated by the decreasing Hellinger distance. The logarithmic decrease of the distance

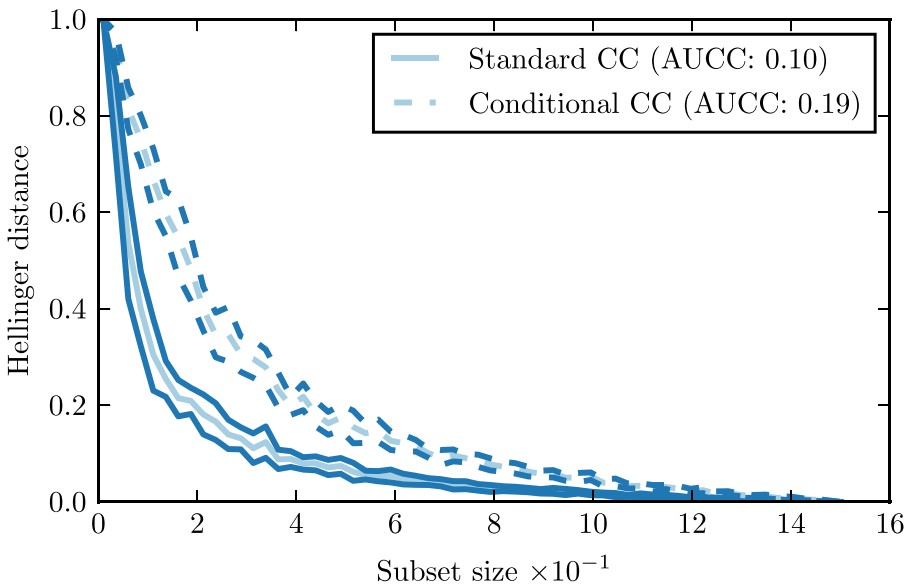

**Figure 1** **Complexity curve (solid) and conditional complexity curve (dashed) for iris data set.**

is characteristic: it means that with a relatively small number of samples we can recover general characteristics of the distribution, but to model the details precisely we need a lot more data points. The shape of the curve is very regular, with just minimal variations. It means that the subset size has a far greater impact on the Hellinger distance that the composition of the individual subsets.

The shape of the complexity curve captures the information on the complexity of the data set. If the data is simple, it is possible to represent it relatively well with just a few instances. In such case, the complexity curve is very steep at the beginning and flattens towards the end of the plot. If the data is complex, the initial steepness of the curve is smaller. That information can be aggregated into a single parameter—the area under the complexity curve (AUCC). If we express the subset size as the fraction of the whole data set, then the value of the area under the curve becomes limited to the range $[0, 1]$ and can be used as an universal measure for comparing complexity of different data sets.

## Conditional complexity curve

The complexity curve methodology presented so far deals with the complexity of a data set as a whole. While this approach gives information about data structure, it may assess complexity of the classification task incorrectly. This is because data distribution inside each of the classes may vary greatly from the overall distribution. For example, when the number of classes is larger, or the classes are imbalanced, a random sample large enough to represent the whole data set may be too small to represent some of the classes. To take this into account, we introduce conditional complexity curve. We calculate it by splitting each data sample according to the class value and taking the arithmetic mean of the complexities of each sub-sample. presents the exact procedure.

---

**Algorithm 2** Procedure for calculating conditional complexity curve.

$D$ – original data set, $C$ – number of classes, $N$ – number of subsets, $K$ – number of samples.

1. Transform $D$ with whitening transform and/or ICA to obtain $D_I$.
2. Split $D_I$ according to the class into $D_I^1, D_I^2, \ldots, D_I^C$.
3. From $D_I^1, D_I^2, \ldots, D_I^C$ estimate probability distributions $P^1, P^2, \ldots, P^C$.
4. For $i$ in $1 \ldots |D_I|$ with a step size $\frac{|D_I|}{N}$:
   (a) For $j$ in $1 \ldots K$:
      i. Draw subset $S_i^j \subseteq D_I$ such that $|S_i^j| = i$.
      ii. Split $S_i^j$ according to the class into $S_i^{j,1}, S_i^{j,2}, \ldots, S_i^{j,C}$.
      iii. From $S_i^{j,1}, S_i^{j,2}, \ldots, S_i^{j,C}$ estimate probability distributions $Q_i^{j,1}, Q_i^{j,2}, \ldots, Q_i^{j,C}$.
      iv. Calculate mean Hellinger distance: $l_i^j = \frac{1}{C} \sum_{k=1}^{C} H^2(P^k, Q_i^{j,k})$.
   (b) Calculate mean $m_i$ and standard error $s_i$:

$$m_i = \frac{1}{K} \sum_{j=1}^{K} l_i^j \qquad s_i = \sqrt{\frac{1}{K} \sum_{j=1}^{K} \left( m_i - l_i^j \right)^2}$$

Conditional complexity curve is a plot of $m_i \pm s_i$ vs $i$.

---

Comparison of standard complexity curve and conditional complexity curve for the **iris** data set is given by Fig. 1. This data set has three distinct classes. Our expectation is that estimating conditional distributions for each class would require larger data samples than estimating the overall distribution. Shape of the conditional complexity curve is consistent with this expectation: it is less steep than the standard curve and has larger AUCC value.

## PROPERTIES

To support validity of the proposed method, we perform an in-depth analysis of its properties. We start from purely mathematical analysis, giving some intuitions on the complexity curve convergence rate and identifying border cases. Then, we perform experiments with toy artificial data sets testing basic assumptions behind complexity curve. After that, we compare it experimentally with other complexity data measures and show its usefulness in explaining classifier performance.

### Mathematical properties

Drawing a random subset $S_n$ from a finite data set $D$ of size $N$ corresponds to sampling without replacement. Let assume that the data set contains $k$ distinct values $\{v_1, v_2, \ldots, v_k\}$ occurring with frequencies $P = (p_1, p_2, \ldots, p_k)$. $Q_n = (q_1, q_2, \ldots, q_k)$ will be a random vector which follows a multivariate hypergeometric distribution.

$$q_i = \frac{1}{n} \sum_{y \in S_n} \mathbf{1}\{y = v_i\}.$$

The expected value for any single element is:

$$E[q_i] = p_i.$$

The probability of obtaining any specific vector of frequencies:

$$P\left(Q_n=(q_1,q_2,\ldots,q_k)\right)=\frac{\binom{p_1N}{q_1n}\binom{p_2N}{q_2n}\cdots\binom{p_kN}{q_kn}}{\binom{N}{n}}$$

with $\sum_{i=1}^{k}q_i=1$.

We will consider the simplest case of discrete probability distribution estimated through frequency counts without using the independence assumption. In such case complexity curve is by definition:

$$CC(n)=E[H^2(P,Q_n)].$$

It is obvious that $CC(N)=0$ because when $n=N$ we draw all available data. This means that complexity curve always converges. We can ask whether it is possible to say anything about the rate of this convergence. This is the question about the upper bound on the tail of hypergeometric distribution. Such bound is given by Hoeffding-Chvátal inequality (*Chvátal, 1979*; *Skala, 2013*). For the univariate case it has the following form:

$$P\left(|q_i-p_i|\ge\delta\right)\le 2e^{-2\delta^2n}$$

which generalises to a multivariate case as:

$$P\left(|Q_n-P|\ge\delta\right)\le 2ke^{-2\delta^2n}$$

where $|Q_n-P|$ is the total variation distance. Since $H^2(P,Q_n)\le|Q_n-P|$ this guarantees that complexity curve converges at least as fast.

Now we will consider a special case when $n=1$. In this situation the multivariate hypergeometric distribution is reduced to a simple categorical distribution $P$. In such case the expected Hellinger distance is:

$$E[H^2(P,Q_1)]=\sum_{i=1}^{k}\frac{p_i}{\sqrt{2}}\sqrt{\sum_{j=1}^{k}\left(\sqrt{p_j}-\mathbf{1}\{j=k\}\right)^2}$$

$$=\sum_{i=1}^{k}\frac{p_i}{\sqrt{2}}\sqrt{1-p_i+\left(\sqrt{p_i}-1\right)^2}=\sum_{i=1}^{k}p_i\sqrt{1-\sqrt{p_i}}.$$

This corresponds to the first point of complexity curve and determines its overall steepness.

**Theorem:** $E[H^2(P,Q_1)]$ is maximal for a given $k$ when $P$ is an uniform categorical distribution over $k$ categories, i.e.,:

$$E[H^2(P,Q_1)]=\sum_{i=1}^{k}p_i\sqrt{1-\sqrt{p_i}}\le\sqrt{1-\sqrt{\frac{1}{k}}}.$$

**Proof:** We will consider an arbitrary distribution $P$ and the expected Hellinger distance $E[H^2(P, Q_1)]$. We can modify this distribution by choosing two states $l$ and $k$ occurring with probabilities $p_l$ and $p_k$ such as that $p_l - p_k$ is maximal among all pairs of states. We will redistribute the probability mass between the two states creating a new distribution $P'$. The expected Hellinger distance for the distribution $P'$ will be:

$$E[H^2(P', Q_1)] = \sum_{i=1, i\neq k, i\neq l}^{k} p_i\sqrt{1-\sqrt{p_i}} + a\sqrt{1-\sqrt{a}} + (p_k+p_l-a)\sqrt{1-\sqrt{p_k+p_l-a}}$$

where $a$ and $p_k + p_l - a$ are new probabilities of the two states in $P'$. We will consider a function $f(a) = a\sqrt{1-\sqrt{a}} + (p_k+p_l-a)\sqrt{1-\sqrt{p_k+p_l}}$ and look for its maxima.

$$\frac{\partial f(x)}{\partial a} = -\sqrt{1-\sqrt{p_k+p_l-a}} + \frac{\sqrt{p_k+p_l-a}}{4\sqrt{1-\sqrt{p_k+p_l-a}}} + \sqrt{1-\sqrt{a}} - \frac{\sqrt{a}}{4\sqrt{1-\sqrt{a}}}.$$

The derivative is equal to 0 if and only if $a = \frac{p_k+p_l}{2}$. We can easily see that:

$$f(0) = f(p_k+p_l) = (p_k+p_l)\sqrt{1-\sqrt{p_k+p_l}} < (p_k+p_l)\sqrt{1-\sqrt{\frac{p_k+p_l}{2}}}.$$

This means that $f(a)$ reaches its maximum for $a = \frac{p_k+p_l}{2}$. From that, we can conclude that for any distribution $P$ if we produce distribution $P'$ by redistributing probability mass between two states equally the following holds:

$$E[H^2(P', Q_1)] \geq E[H^2(P, Q_1)].$$

If we repeat such redistribution arbitrary number of times the outcome distribution converges to uniform distribution. This proves that the uniform distribution leads to the maximal expected Hellinger distance for a given number of states.

**Theorem:** Increasing the number of categories by dividing an existing category into two new categories always increases the expected Hellinger distance, i.e.,

$$\sum_{i=1}^{k} p_i\sqrt{1-\sqrt{p_i}} \leq \sum_{i=1, i\neq l}^{k} p_i\sqrt{1-\sqrt{p_i}} + a\sqrt{1-\sqrt{a}} + (p_l-a)\sqrt{1-\sqrt{p_l-a}}.$$

**Proof:** Without the loss of generality, we can assume that $a < 0.5 p_l$. We can subtract terms occurring on both sides of the inequality obtaining:

$$p_l\sqrt{1-\sqrt{p_l}} \leq a\sqrt{1-\sqrt{a}} + (p_l-a)\sqrt{1-\sqrt{p_l-a}}$$

$$p_l\sqrt{1-\sqrt{p_l}} \leq a\sqrt{1-\sqrt{a}} + p_l\sqrt{1-\sqrt{p_l-a}} - a\sqrt{1-\sqrt{p_l-a}}$$

$$p_l\sqrt{1-\sqrt{p_l}} + a\sqrt{1-\sqrt{p_l-a}} \leq a\sqrt{1-\sqrt{a}} + p_l\sqrt{1-\sqrt{p_l-a}}.$$

Now we can see that:

$$p_l\sqrt{1-\sqrt{p_l}} \leq p_l\sqrt{1-\sqrt{p_l-a}}$$

and

$$a\sqrt{1-\sqrt{p_l-a}} \leq a\sqrt{1-\sqrt{a}}$$

which concludes the proof.

From the properties stated by these two theorems, we can gain some intuitions about complexity curves in general. First, by looking at the formula for the uniform distribution $E[H^2(P,Q_1)] = \sqrt{1-\sqrt{\frac{1}{k}}}$ we can see that when $k=1$ $E[H^2(P,Q_1)] = 0$ and when $k \to \infty$ $E[H^2(P,Q_1)] \to 1$. The complexity curve will be less steep if the variables in the data set take multiple values and each value occurs with equal probability. This is consistent with our intuition: we need a larger sample to cover such space and collect information. For a smaller number of distinct values or distributions with mass concentrated mostly in a few points, a smaller sample will be sufficient to represent most of the information in the data set.

## Complexity curve and the performance of an unbiased model

To confirm validity of the assumptions behind the complexity curve, we performed experiments with artificial data generated according to known models. For each of the data set, we selected an appropriate classifier which is known to be unbiased with respect to the given model. In this way it was possible to observe if the variance error component is indeed upper bounded by the complexity curve. To train the classifiers, we used the same setting as when calculating the complexity curve: classifiers were trained on random subsets and tested on the whole data set. We fitted the learning curve to the complexity curve by matching first and last points of both curves. We then observed the relation of the two curves in between.

The first generated data set followed the logistic model (**logit** data set). Matrix $X$ (1,000 observations, 12 attributes) contained values drawn from the normal distribution with mean 0 and standard deviation 1. Class vector $Y$ was defined as follows:

$$P(Y|x) = \frac{e^{\beta'x}}{\left(1+e^{\beta'x}\right)}$$

where $\beta = (0.2, 0.3, 0.4, 0.5, 0.6, 0.7, 0, 0, 0, 0, 0, 0)$. All attributes were independent and conditionally independent. Since $Y$ values were determined in a non-deterministic way, there was some noise present –classification error of the logistic regression classifier trained and tested on the full data set was larger than zero.

Figure 2 presents the complexity curve and the adjusted error of logistic regression for the generated data. After ignoring the noise error component, we can see that the variance error component is indeed upper bounded by the complexity curve.

Different kind of artificial data represented multidimensional space with parallel stripes in one dimension (**stripes** data set). It consisted of $X$ matrix with 1,000 observations and 10 attributes drawn from an uniform distribution defined on the range $[0,1)$. Class values $Y$ depended only on the values of one of the attributes: for values lesser than 0.25 or greater than 0.75 the class was 1, for other values the class was 0. This kind of relation can

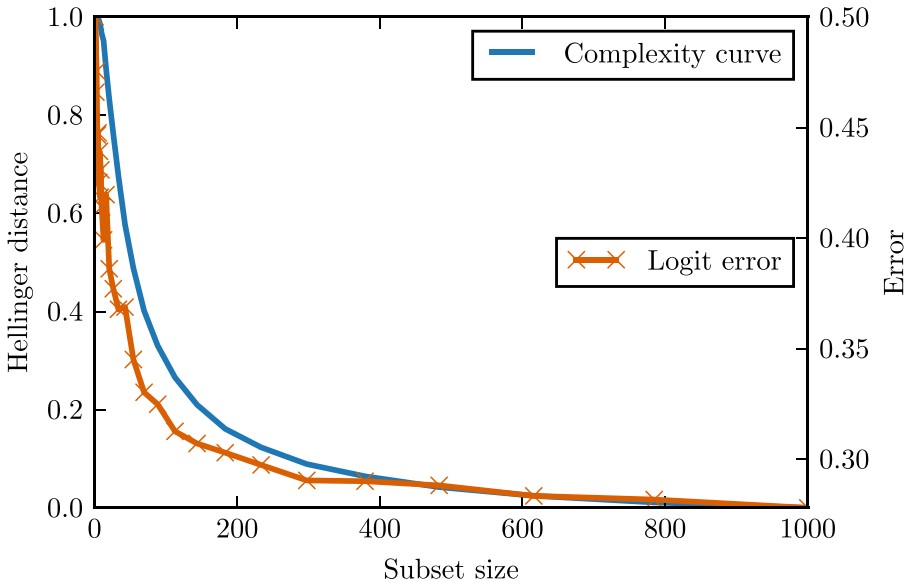

**Figure 2** Complexity curve and learning curve of the logistic regression on the logit data.

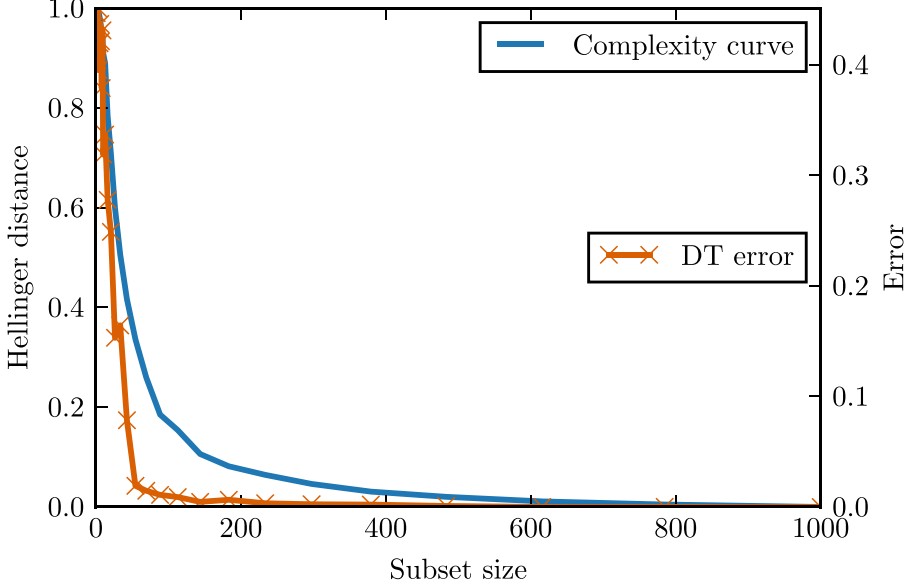

**Figure 3** Complexity curve and learning curve of the decision tree on the stripes data.

be naturally modelled with a decision tree. All the attributes are again independent and conditionally independent.

Figure 3 presents complexity curve and the adjusted error of decision tree classifier on the generated data. Once again the assumptions of complexity curve methodology are satisfied and the complexity curve indeed an upper bounds the classification error.

What would happen if the attribute conditional independence assumption was broken? To answer this question, we generated another type of data modelled after multidimensional

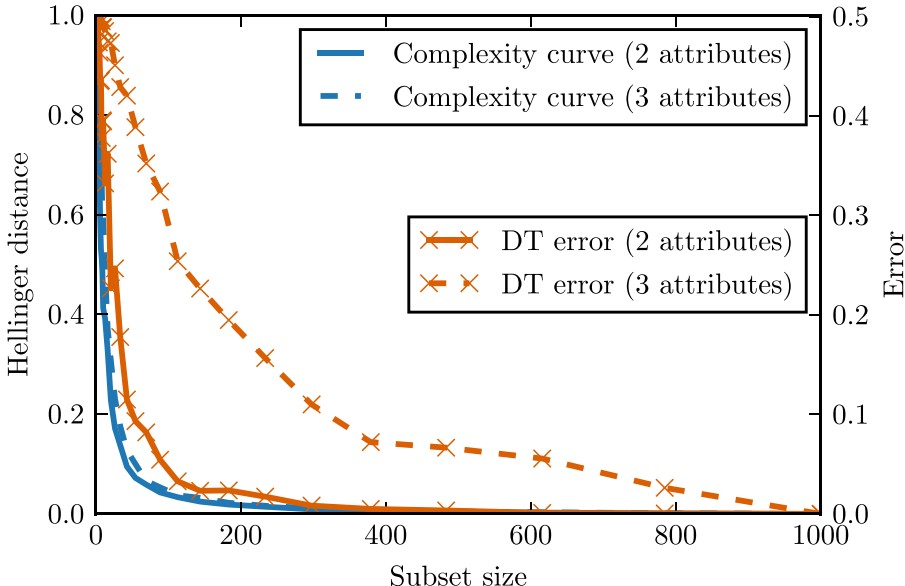

**Figure 4** Complexity curve and learning curve of the decision tree on the chessboard data.

chessboard (**chessboard** data set). $X$ matrix contained 1,000 observations and 2, 3 attributes drawn from an uniform distribution on range $[0, 1]$. Class vector $Y$ had the following values:

$$\begin{cases} 0 & \text{if } \Sigma_{i=0}^{m} \left\lfloor \dfrac{x_i}{s} \right\rfloor \text{ is even} \\ 1 & \text{otherwise} \end{cases}$$

where $s$ was a grid step in our experiments set to 0.5. There is clearly strong attribute dependence, but since all parts of decision boundary are parallel to one of the attributes this kind of data can be modelled with a decision tree with no bias.

Figure 4 presents complexity curves and error curves for different dimensionalities of **chessboard** data. Here the classification error becomes larger than indicated by complexity curve. The more dimensions, the more dependencies between attributes violating complexity curve assumptions. For a three-dimensional chessboard the classification problem becomes rather hard and the observed error decreases slowly, but the complexity curve remains almost the same as for a two-dimensional case. This shows that the complexity curve is not expected to be a good predictor of classification accuracy in the problems where a lot of high-dimensional attribute dependencies occur for example, in epistatic domains in which the importance of one attribute depends on the values of the other.

The results of experiments with controlled artificial data sets are consistent with our theoretical expectations. Based on these results, we can introduce a general interpretation of the difference between complexity curve and learning curve: learning curve below the complexity curve is an indication that the algorithm is able to build a good model without sampling the whole domain, limiting the variance error component. On the other hand, the learning curve above the complexity curve is an indication that the algorithm includes complex attributes dependencies in the constructed model, promoting the variance error component.

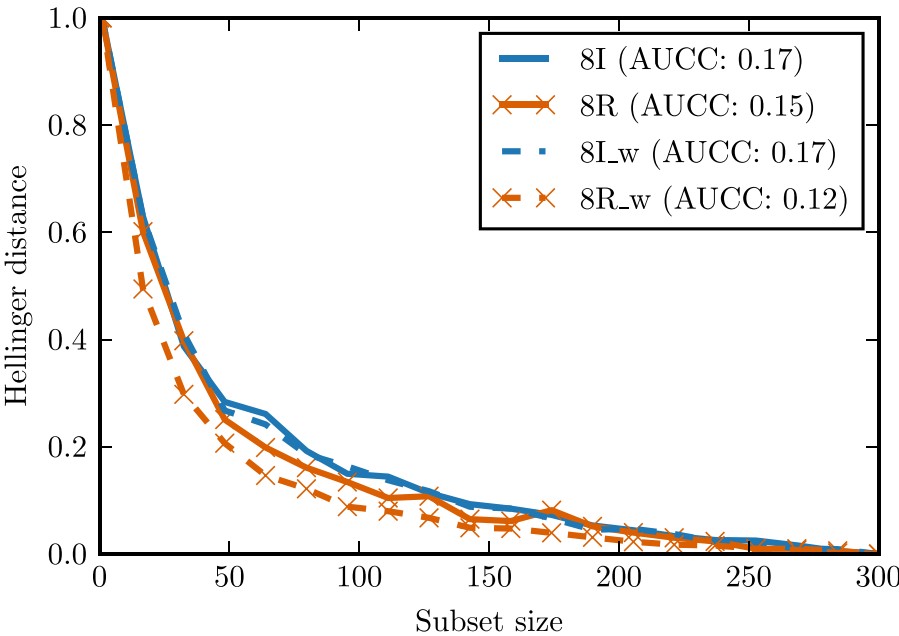

**Figure 5** **Complexity curves for whitened data (dashed lines) and not whitened data (solid lines).** Areas under the curves are given in the legend. 8I—set of 8 independent random variables with Student's t distribution. 8R—one random variable with Student's t distribution repeated 8 times. 8I_w—whitened 8I. 8R_w—whitened 8R.

## Impact of whitening and ICA

To evaluate the impact of the proposed preprocessing techniques (whitening and ICA—Independent Component Analysis) on complexity curves, we performed experiments with artificial data. In the first experiment, we generated two data sets of 300 observations and with eight attributes distributed according to Student's t distribution with 1.5 degrees of freedom. In one data set all attributes were independent, in the other the same attribute was repeated eight times. Small Gaussian noise was added to both sets. Figure 5 shows complexity curves calculated before and after whitening transform. We can see that whitening had no significant effect on the complexity curve of the independent set. In the case of the dependent set, complexity curve calculated after whitening decreases visibly faster and the area under the curve is smaller. This is consistent with our intuitive notion of complexity: a data set with highly correlated or duplicated attributes should be significantly less complex.

In the second experiment, two data sets with 100 observations and four attributes were generated. The first data set was generated from the continuous uniform distribution on interval [0, 2], the second one from the discrete (categorical) uniform distribution on the same interval. Small Gaussian noise was added to both sets. Figure 6 presents complexity curves for original, whitened and ICA-transformed data. Among the original data sets, the intuitive notion of complexity is preserved: the area under the complexity curve for categorical data is smaller. The difference disappears for the whitened data but is again visible in the ICA-transformed data.

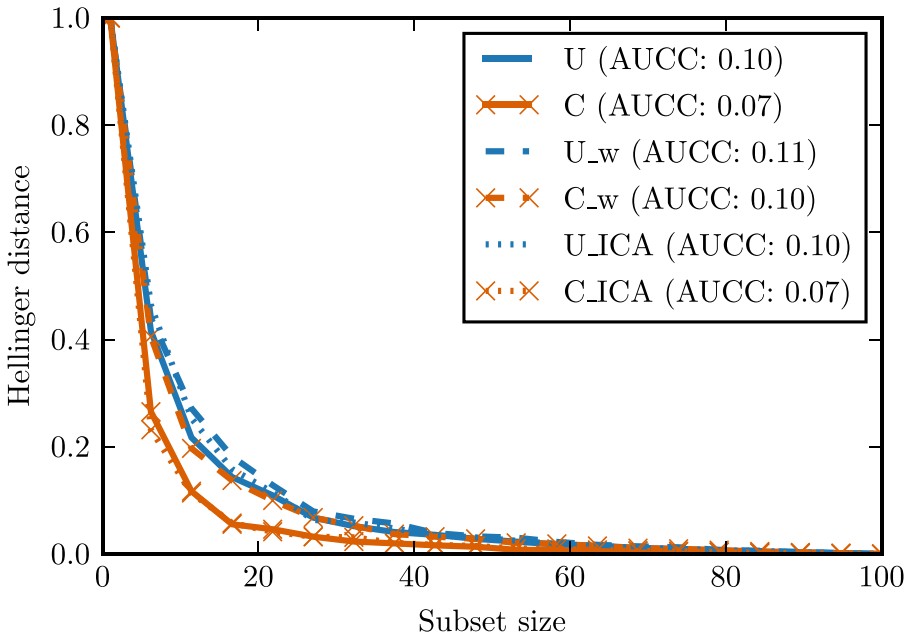

**Figure 6** **Complexity curves for whitened data (dashed lines), not whitened data (solid lines) and ICA-transformed data (dotted lines).** Areas under the curves are given in the legend. U—data sampled from uniform distribution. C—data sampled from categorical distribution. U_w—whitened U. C_w—whitened C. U_ICA—U_w after ICA. C_ICA—C_w after ICA.

These simple experiments are by no means exhaustive but they confirm usefulness of the chosen signal processing techniques (data whitening and the Independent Component Analysis) in the complexity curve analysis.

## Complexity curve variability and outliers

The complexity curve is based on the expected Hellinger distance and the estimation procedure includes some variance. The natural assumption is that the variability caused by the sample size is greater than the variability resulting from a specific composition of a sample. Otherwise, averaging over samples of the same size would not be meaningful. This assumption is already present in standard learning curve methodology where classifier accuracy is plotted against training set size. We expect that the exact variability of the complexity curve will be connected with the presence of outliers in the data set. Such influential observations will have a huge impact depending on whether they will be included in a sample or not.

To verify whether these intuitions were true, we constructed two new data sets by introducing artificially outliers to WINE data set. In WINE001 we modified 1% of the attribute values by multiplying them by a random number from range $(-10, 10)$. In WINE005 5% of the values were modified in such manner.

Figure 7 presents conditional complexity curves for all three data sets. WINE001 curve has indeed a higher variance and is less regular than WINE curve. WINE005 curve is characterised not only by a higher variance but also by a larger AUCC value. This means that adding so much noise increased the overall complexity of the data set significantly.

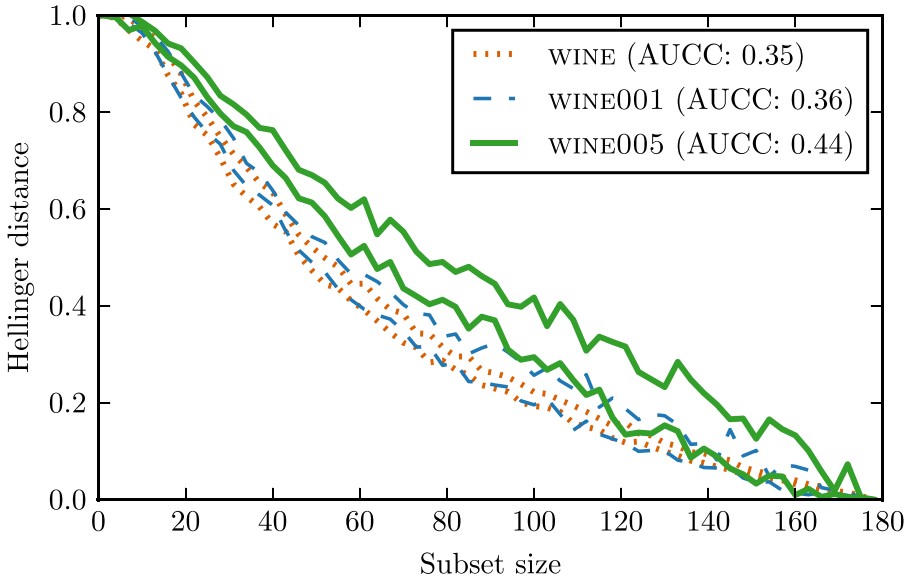

**Figure 7** **Complexity curves for WINE and its counterparts with introduced outliers.** For the sake of clarity only contours were drawn.

The result support our hypothesis that large variability of complexity curve signify an occurrence of highly influential observations in the data set. This makes complexity curve a valuable diagnostic tool for such situations. However, it should be noted that our method is unable to distinguish between important outliers and plain noise. To obtain this kind of insight, one has to employ different methods.

## Comparison with other complexity measures

The set of data complexity measures developed by *Ho & Basu (2002)* and extended by *Ho, Basu & Law (2006)* continues to be used in experimental studies to explain performance of various classifiers (*Diez-Pastor et al., 2015*; *Mantovani et al., 2015*). We decided to compare experimentally complexity curve with those measures. Descriptions of the measures used are given in Table 1.

According to our hypothesis conditional complexity curve should be robust in the context of class imbalance. To demonstrate this property, we used for the comparison 81 imbalanced data sets used previously in the study by *Diez-Pastor et al. (2015)*. These data sets come originally from HDDT (*Cieslak et al., 2011*) and KEEL (*Alcalá et al., 2011*) repositories. We selected only binary classification problems. The list of data sets with their properties is presented in Supplemental Information 1 as Table S1 and Table S2.

For each data set, we calculated the area under the complexity curve using the previously described procedure and the values of other data complexity measures using DCOL software (*Orriols-Puig, Macià & Ho, 2010*). Pearson's correlation was then calculated for all the measures. As the T2 measure seemed to have non-linear characteristics destroying the correlation additional column logT2 was added to comparison. Results are presented in Fig. 8. Clearly, AUCC is mostly correlated with logT2 measure. This is to be expected as both measures are concerned with sample size in relation to attribute structure. The

**Table 1  Data complexity measures used in experiments.**

| Id | Description |
|---|---|
| F1 | Maximum Fisher's discriminant ratio |
| F1v | Directional-vector maximum Fisher's discriminant ratio |
| F2 | Overlap of the per-class bounding boxes |
| F3 | Maximum individual feature efficiency |
| F4 | Collective feature efficiency |
| L1 | Minimized sum of the error distance of a linear classifier |
| L2 | Training error of a linear classifier |
| L3 | Nonlinearity of a linear classifier |
| N1 | Fraction of points on the class boundary |
| N2 | Ratio of average intra/inter class nearest neighbour distance |
| N3 | Leave-one-out error rate of the one-nearest neighbour classifier |
| N4 | Nonlinearity of the one-nearest neighbour classifier |
| T1 | Fraction of maximum covering spheres |
| T2 | Average number of points per dimension |

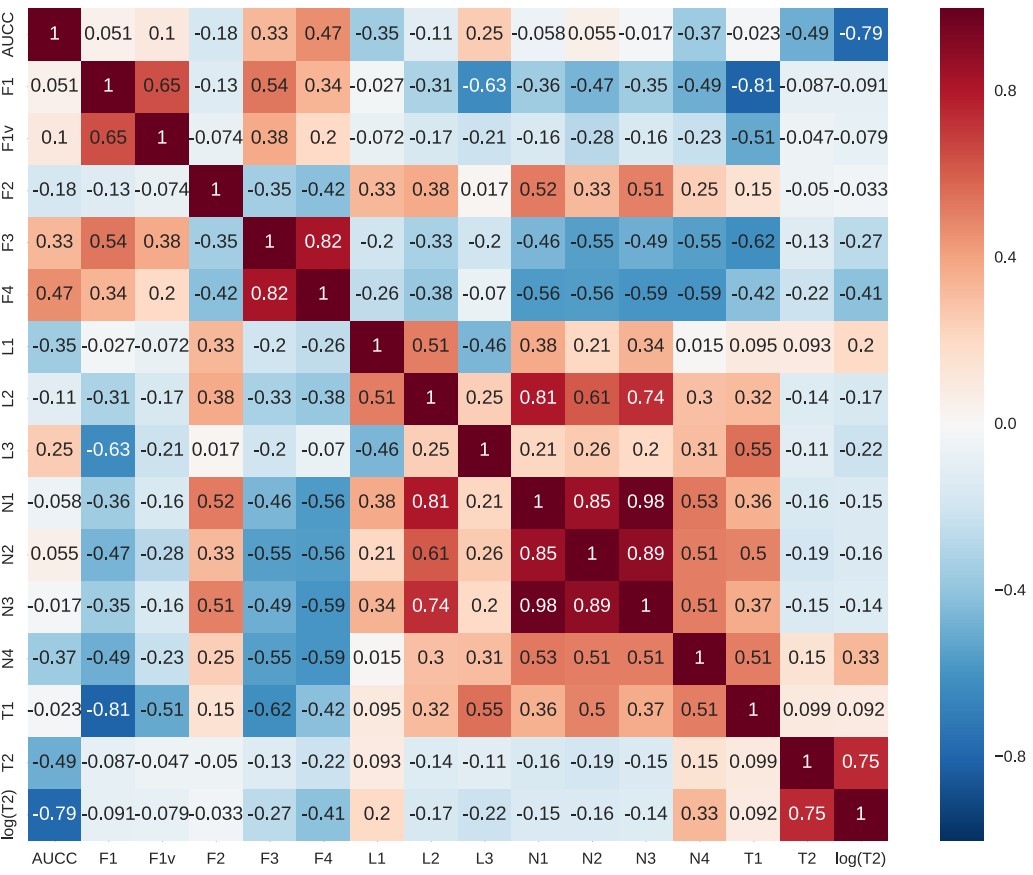

**Figure 8   Pearson's correlations between complexity measures.**

**Table 2  Pearson's correlations coefficients between classifier AUC ROC performances and complexity measures.** The largest absolute value in each row is printed in bold.

| | AUCC | logT2 |
| --- | --- | --- |
| LDA | 0.0489 | 0.0227 |
| Logistic regression | −0.0539 | **0.1103** |
| Naive Bayes | −0.0792 | **0.0889** |
| 1-NN | **−0.1256** | 0.0772 |
| 3-NN | **−0.1311** | 0.0863 |
| 5-NN | **−0.1275** | 0.0952 |
| 10-NN | **−0.1470** | 0.1225 |
| 15-NN | **−0.1730** | 0.1584 |
| 20-NN | **−0.1842** | 0.1816 |
| 25-NN | −0.1859 | **0.1902** |
| 30-NN | −0.1969 | **0.2059** |
| 35-NN | −0.2249 | **0.2395** |
| Decision tree $d = 1$ | 0.0011 | **−0.0624** |
| Decision tree $d = 3$ | **−0.1472** | 0.1253 |
| Decision tree $d = 5$ | −0.1670 | **0.1690** |
| Decision tree $d = 10$ | **−0.1035** | 0.0695 |
| Decision tree $d = 15$ | **−0.0995** | 0.0375 |
| Decision tree $d = 20$ | **−0.0921** | 0.0394 |
| Decision tree $d = 25$ | **−0.0757** | 0.0298 |
| Decision tree $d = 30$ | **−0.0677** | 0.0227 |
| Decision tree $d = \inf$ | **−0.0774** | 0.0345 |

difference is that T2 takes into account only the number of attributes while AUCC considers also the shape of distributions of the individual attributes. Correlations of AUCC with other measures are much lower and it can be assumed that they capture different aspects of data complexity and may be potentially complementary.

The next step was to show that information captured by AUCC is useful for explaining classifier performance. In order to do so, we trained a number of different classifiers on the 81 benchmark data sets and evaluated their performance using random train-test split with proportion 0.5 repeated 10 times. The performance measure used was the area under the ROC curve. We selected three linear classifiers—naïve Bayes with Gaussian kernel, linear discriminant analysis (LDA) and logistic regression—and two families of non-linear classifiers of varying complexity: $k$-nearest neighbour classifier (k-NN) with different values of parameter $k$ and decision tree (CART) with the limit on maximal tree depth. The intuition was as follows: the linear classifiers do not model attributes interdependencies, which is in line with complexity curve assumptions. Selected non-linear classifiers on the other hand are—depending on the parametrisation—more prone to variance error, which should be captured by the complexity curve.

Correlations between AUCC, logT2, and classifier performance are presented in Table 2. Most of the correlations are weak and do not reach statistical significance; however, some general tendencies can be observed. As can be seen, AUC ROC scores of linear

classifiers have very little correlation with AUCC and logT2. This may be explained by the high-bias and low-variance nature of these classifiers: they are not strongly affected by data scarcity but their performance depends on other factors. This is especially true for the LDA classifier, which has the weakest correlation among linear classifiers.

In $k$-NN, classifier complexity depends on $k$ parameter: with low $k$ values, it is more prone to variance error, with a larger $k$ it is prone to bias if the sample size is not large enough (*Domingos, 2000*). Both AUCC and logT2 seem to capture the effect of sample size in the case of large $k$ values well (correlations $-0.2249$ and $0.2395$ for 35-NN). However, for $k = 1$ the correlation with AUCC is stronger ($-0.1256$ vs $0.0772$).

Depth parameter in decision tree also regulates complexity: the larger the depth the more classifier is prone to variance error and less to bias error. This suggests that AUCC should be more strongly correlated with performance of deeper trees. On the other hand, complex decision trees explicitly model attribute interdependencies ignored by complexity curve, which may weaken the correlation. This is observed in the obtained results: for a decision stub (tree of depth 1), which is low-variance high-bias classifier, correlation with AUCC and logT2 is very weak. For $d = 3$ and $d = 5$ it becomes visibly stronger, and then for larger tree depth it again decreases. It should be noted that with large tree depth, as with small $k$ values in $k$-NN, AUCC has stronger correlation with the classifier performance than logT2.

A slightly more sophisticated way of applying data complexity measures is an attempt to explain classifier performance relative to some other classification method. In our experiments, LDA is a good candidate for reference method since it is simple, has low variance and is not correlated with either AUCC or logT2. Table 3 presents correlations of both measures with classifier performance relative to LDA. Here we can see that correlations for AUCC are generally higher than for logT2 and reach significance for the majority of classifiers. Especially in the case of decision tree, AUCC explains relative performance better than logT2 (correlation $0.1809$ vs $-0.0303$ for $d = \text{inf}$).

Results of the presented correlation analyses demonstrate the potential of the complexity curve to complement the existing complexity measures in explaining classifier performance. As expected from theoretical considerations, there is a relation between how well AUCC correlates with classifier performance and the classifier's position in the bias–variance spectrum. It is worth noting that despite the attribute independence assumption the complexity curve method proved useful for explaining performance of complex non-linear classifiers.

## Large $p$, small $n$ problems

There is a special category of machine learning problems in which the number of attributes $p$ is large with respect to the number of samples $n$, perhaps even order of magnitudes larger. Many important biological data sets, most notably data from microarray experiments, fall into this category (*Johnstone & Titterington, 2009*). To test how our complexity measure behaves in such situations, we calculated AUCC scores for a few microarray data sets and compared them with AUC ROC scores of some simple classifiers. Classifiers were evaluated as in the previous section. Detailed information about the data sets is given in Supplemental Information 1 as Table S3.

**Table 3  Pearson's correlations coefficients between classifier AUC ROC performances relative to LDA performance and complexity measures.** The largest absolute value in each row is printed in bold.

|  | AUCC | logT2 |
|---|---|---|
| LDA - Logistic regression | **0.2026** | −0.2025 |
| LDA - Naive Bayes | **0.2039** | −0.1219 |
| LDA - 1-NN | **0.2278** | −0.0893 |
| LDA - 3-NN | **0.2482** | −0.1063 |
| LDA - 5-NN | **0.2490** | −0.1210 |
| LDA - 10-NN | **0.2793** | −0.1609 |
| LDA - 15-NN | **0.3188** | −0.2148 |
| LDA - 20-NN | **0.3365** | −0.2510 |
| LDA - 25-NN | **0.3392** | −0.2646 |
| LDA - 30-NN | **0.3534** | −0.2868 |
| LDA - 35-NN | **0.3798** | −0.3259 |
| LDA - Decision tree $d = 1$ | 0.0516 | **0.1122** |
| LDA - Decision tree $d = 3$ | **0.3209** | −0.1852 |
| LDA - Decision tree $d = 5$ | **0.3184** | −0.2362 |
| LDA - Decision tree $d = 10$ | **0.2175** | −0.0838 |
| LDA - Decision tree $d = 15$ | **0.2146** | −0.0356 |
| LDA - Decision tree $d = 20$ | **0.2042** | −0.0382 |
| LDA - Decision tree $d = 25$ | **0.1795** | −0.0231 |
| LDA - Decision tree $d = 30$ | **0.1636** | −0.0112 |
| LDA - Decision tree $d = \inf$ | **0.1809** | −0.0303 |

Results of the experiment are presented in Table 4. As expected, with the number of attributes much larger than the number of observations, data is considered by our metric as extremely scarce –values of AUCC are in all cases above 0.95. On the other hand, the AUC ROC classification performance is very varied between data sets with scores approaching or equal to 1.0 for Leukemia and Lymphoma data sets, and scores around 0.5 baseline for Prostate. This is because, despite the large number of dimensions, the form of the optimal decision function can be very simple, utilising only a few of available dimensions. The complexity curve does not consider the shape of decision boundary at all, and thus does not reflect differences in classification performance.

From this analysis we concluded that complexity curve is not a good predictor of classifier performance for data sets containing a large number of redundant attributes, as it does not differentiate between important and unimportant attributes. The logical way to proceed in such case would be to perform some form of feature selection or dimensionality reduction on the original data, and then calculate the complexity curve in the reduced dimensions.

## APPLICATIONS

### Interpreting complexity curves

In order to prove the practical applicability of the proposed methodology, and show how complexity curve plot can be interpreted, we performed experiments with six simple

**Table 4 Areas under conditional complexity curve (AUCC) for microarray data sets along AUC ROC values for different classifiers.**

| Dataset | AUCC | 1-NN | 5-NN | DT d-10 | DT d-inf | LDA | NB | LR |
|---|---|---|---|---|---|---|---|---|
| Adenocarcinoma | 0.9621 | 0.6354 | 0.5542 | 0.5484 | 0.5172 | 0.6995 | 0.5021 | 0.7206 |
| Breast2 | 0.9822 | 0.5869 | 0.6572 | 0.6012 | 0.6032 | 0.6612 | 0.5785 | 0.6947 |
| Breast3 | 0.9830 | 0.6788 | 0.7344 | 0.6274 | 0.6131 | 0.7684 | 0.6840 | 0.7490 |
| Colon | 0.9723 | 0.7395 | 0.7870 | 0.6814 | 0.6793 | 0.7968 | 0.5495 | 0.8336 |
| Leukemia | 0.9611 | 1.0000 | 0.9985 | 0.7808 | 0.8715 | 0.9615 | 0.8300 | 1.0000 |
| Lymphoma | 0.9781 | 0.9786 | 0.9976 | 0.8498 | 0.8660 | 0.9952 | 0.9700 | 1.0000 |
| Prostate | 0.9584 | 0.5931 | 0.4700 | 0.4969 | 0.5238 | 0.4908 | 0.5000 | 0.4615 |

**Notes.**

$k$-NN, $k$-nearest neighbour; DT, CART decision tree; LDA, linear discriminant analysis; NB, naïve Bayes; LR, logistic regression.

data sets from UCI Machine Learning Repository (*Frank & Asuncion, 2010*). The sets were chosen only as illustrative examples. The basic properties of the data sets are given in Supplemental Information as Table S4. For each data set, we calculated conditional complexity curve. The curves are presented in Fig. 9. Learning curves of CART decision tree (DT) were included for comparison.

On most of the benchmark data sets we can see that complexity curve upper bounds the DT learning curve. The bound is relatively tight in the case of GLASS and IRIS, and looser for BREAST-CANCER-WISCONSIN and WINE data set. A natural conclusion is that a lot of variability contained in this last data set and captured by the Hellinger distance is irrelevant to the classification task. The most straightforward explanation would be the presence of unnecessary attributes not correlated with the class which can be ignored altogether. This is consistent with the results of various studies in feature selection. *Choubey et al. (1996)* identified that in GLASS data 7–8 attributes (78–89%) are relevant, in IRIS data 3 attributes (75%), and in BREAST-CANCER-WISCONSIN 5–7 attributes (56–78%). Similar results were obtained for BREAST-CANCER-WISCONSIN in other studies, which found that only 4 of the original attributes (44%) contribute to the classification (*Ratanamahatana & Gunopulos, 2003*; *Liu, Motoda & Dash, 1998*). *Dy & Brodley (2004)* obtained best classification results for WINE data set with 7 attributes (54%).

On MONKS-1 and CAR, the complexity curve is no longer a proper upper bound on the DT learning curve. This is an indication of models relying heavily on attribute interdependencies to determine the correct class. This is not surprising: both MONKS-1 and CAR are artificial data sets with discrete attributes devised for evaluation of rule-based and tree-based classifiers (*Thrun et al., 1991*; *Bohanec & Rajkovič, 1988*). Classes are defined with logical formulas utilising relations of multiple attributes rather than single values—clearly the attributes are interdependent. In that context, the complexity curve can be treated as a baseline for independent attribute situation and the generalisation curve as diagnostic tool indicating the presence of interdependencies.

Besides the slope of the complexity curve we can also analyse its variability. We can see that the shape of WINE complexity curve is very regular with small variance in each point, while the GLASS curve displays much higher variance. This mean that the observations in

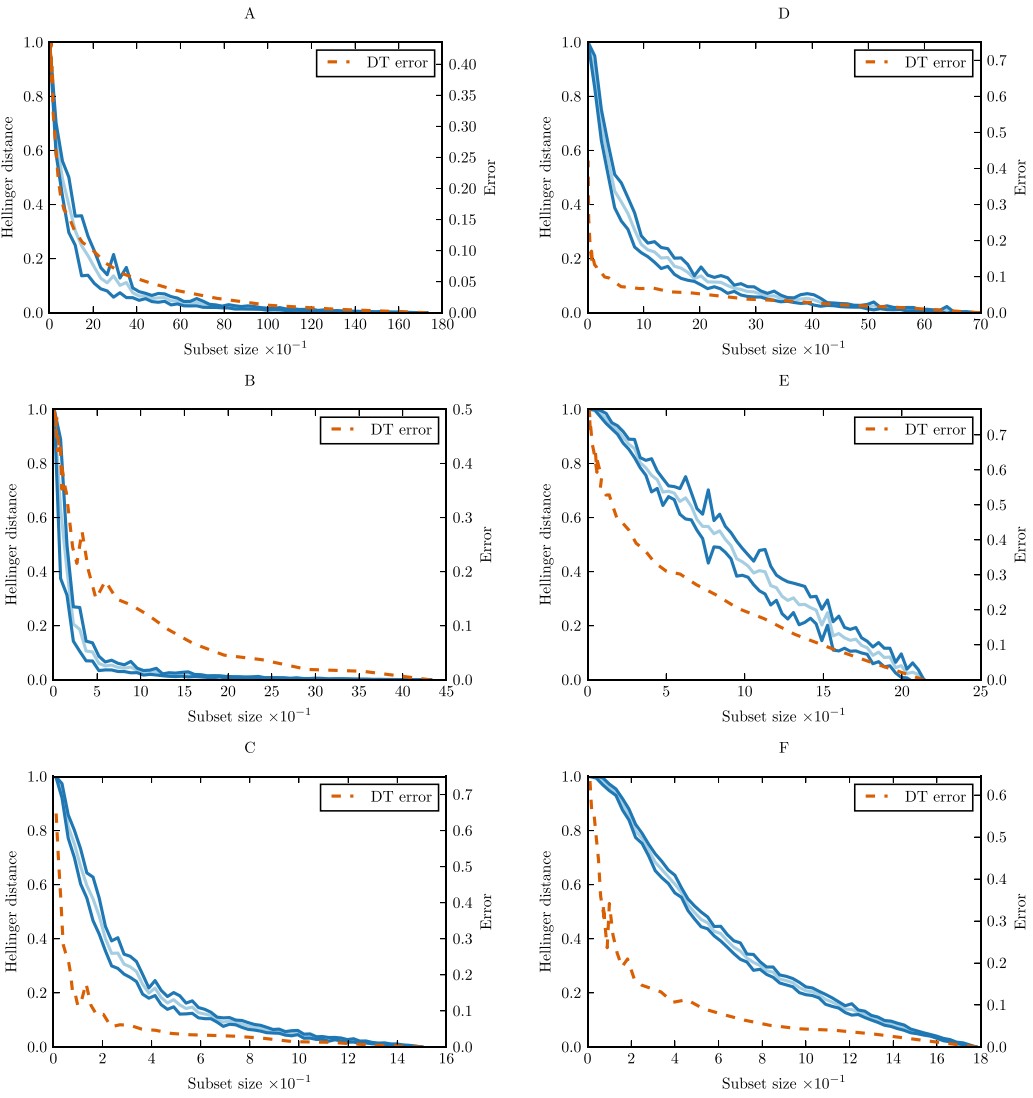

**Figure 9** Conditional complexity curves for six different data sets from UCI Machine Learning repository with areas under complexity curve (AUCC) reported: (A) CAR, AUCC: 0.08, (B) MONKS-1, AUCC: 0.05, (C) IRIS, AUCC: 0.19, (D) BREAST-CANCER-WISCONSIN, AUCC: 0.13, (E) GLASS, AUCC: 0.44, (F) WINE, AUCC: 0.35.

GLASS data set are more diverse and some observations (or their combinations) are more important for representing data structure than the other.

This short analysis demonstrate how to use complexity curves to compare properties of different data sets. Here only decision tree was used as reference classifier. The method can be easily extended to include multiple classifiers and compare their performance. We present such an extended analysis in Supplemental Information 2.

## Data pruning with complexity curves

The problem of data pruning in the context of machine learning is defined as reducing the size of training sample in order to reduce classifier training time and still achieve

satisfactory performance. It becomes extremely important as the data grows and (a) does not fit the memory of a single machine, (b) training times of more complex algorithms become very long.

A classic method for performing data pruning is progressive sampling—training the classifier on data samples of increasing size as long as its performance increases. *Provost, Jensen & Oates (1999)* analysed various schedules for progressive sampling and recommended geometric sampling, in which sample size is multiplied by a specified constant in each iteration, as the reasonable strategy in most cases. Geometric sampling uses samples of sizes $a^i n_0$, where $n_0$—initial sample size, $a$—multiplier, $i$—iteration number.

In our method, instead of training classifier on the drawn data sample, we are probing the complexity curve. We are not trying to detect the convergence of classifier accuracy, but just search for a point on the curve corresponding to some reasonably small Hellinger distance value, e.g., 0.005. This point designates the smallest data subset which still contains the required amount of information.

In this setting, we were not interested in calculating the whole complexity curve but just in finding the minimal data subset, which still contains most of the original information. The search procedure should be as fast as possible, since the goal of the data pruning is to save time spent on training classifiers. To comply with these requirements, we constructed a criterion function of the form $f(x) = H^2(G_x, D) - t$, where $D$ denotes a probability distribution induced by the whole data set, $G_x$ a distribution induced by random subset of size $x$ and $t$ is the desired Hellinger distance. We used classic Brent method (*Brent, 1973*) to find a root of the criterion function. In this way, data complexity was calculated only for the points visited by Brent's algorithm. To speed up the procedure even further, we used the standard complexity curve instead of the conditional one and settled for whitening transform as the only preprocessing technique.

To verify if this idea is of practical use, we performed an experiment with three bigger data sets from UCI repository. Their basic properties are given in Supplemental Information 1 as Table S5.

For all data sets, we performed a stratified 10 fold cross validation experiment. The training part of a split was pruned according to our criterion function with $t = 0.005$ (CC pruning) or using geometric progressive sampling with multiplier $a = 2$ and initial sample size $n_0 = 100$ (PS pruning). Achieving the same accuracy as with CC pruning was used as a stop criterion for progressive sampling. Classifiers were trained on pruned and unpruned data and evaluated on the testing part of each cross validation split. Standard error was calculated for the obtained values. We have used machine learning algorithms from the scikit-learn library (*Pedregosa et al., 2011*) and the rest of the procedure was implemented in Python with the help of NumPy and SciPy libraries. Calculations were done on a workstation with 8 core Intel® Core™ i7-4770 3.4 Ghz CPU working under Arch GNU/Linux.

Table 5 presents measured times and obtained accuracies. As can be seen, the difference in classification accuracies between pruned and unpruned training data is negligible. CC compression rate differs for the three data sets, which suggests that they are of different complexity: for LED data only 5% is needed to perform successful classification, while

**Table 5** **Obtained accuracies and training times of different classification algorithms on unpruned and pruned data sets.** Score corresponds to classifier accuracy, time to classifier training time (including pruning procedure), rate to compression rate. CC corresponds to data pruning with complexity curves, PS to data pruning with progressive sampling.

| Classifier | Score | CC score | Time | CC time | PS time | PS rate |
|---|---|---|---|---|---|---|
| **Waveform** | | | | | | |
| Mean CC compression rate: 0.19 ± 0.02 Mean CC compression time: 4.01 ± 0.14 | | | | | | |
| Linear SVC | 0.86 ± 0.00 | 0.86 ± 0.00 | 27.71 ± 0.35 | **6.69 ± 0.52** | 10.73 ± 8.65 | 0.55 ± 0.49 |
| Gaussian NB | 0.80 ± 0.01 | 0.80 ± 0.01 | **0.02 ± 0.00** | 4.02 ± 0.14 | 0.03 ± 0.01 | 0.01 ± 0.00 |
| RF | 0.86 ± 0.00 | 0.85 ± 0.00 | 33.49 ± 0.04 | **9.29 ± 0.76** | 18.06 ± 10.75 | 0.46 ± 0.37 |
| SVC | 0.86 ± 0.00 | 0.86 ± 0.00 | 211.98 ± 0.93 | **9.08 ± 1.21** | 21.22 ± 28.34 | 0.33 ± 0.42 |
| Tree | 0.78 ± 0.00 | 0.77 ± 0.00 | 3.06 ± 0.06 | 4.50 ± 0.20 | **1.40 ± 0.70** | 0.37 ± 0.28 |
| Logit | 0.86 ± 0.00 | 0.86 ± 0.00 | 1.75 ± 0.06 | 4.21 ± 0.17 | **0.60 ± 0.62** | 0.30 ± 0.34 |
| GBC | 0.86 ± 0.00 | 0.86 ± 0.00 | 112.34 ± 0.12 | **24.59 ± 2.30** | 66.66 ± 37.99 | 0.53 ± 0.43 |
| **Led** | | | | | | |
| Mean CC compression rate: 0.04 ± 0.01 Mean CC compression time: 1.38 ± 0.03 | | | | | | |
| Linear SVC | 0.74 ± 0.00 | 0.74 ± 0.00 | 4.68 ± 0.10 | 1.49 ± 0.04 | **0.47 ± 1.04** | 0.13 ± 0.34 |
| Gaussian NB | 0.74 ± 0.00 | 0.73 ± 0.00 | **0.02 ± 0.00** | 1.38 ± 0.03 | 0.07 ± 0.02 | 0.26 ± 0.44 |
| RF | 0.74 ± 0.00 | 0.73 ± 0.00 | 1.77 ± 0.01 | 1.47 ± 0.03 | **0.83 ± 0.25** | 0.05 ± 0.04 |
| SVC | 0.74 ± 0.00 | 0.74 ± 0.00 | 82.16 ± 0.86 | **1.56 ± 0.07** | 10.04 ± 17.52 | 0.26 ± 0.44 |
| Tree | 0.74 ± 0.00 | 0.73 ± 0.00 | **0.03 ± 0.00** | 1.38 ± 0.03 | 0.04 ± 0.01 | 0.09 ± 0.10 |
| Logit | 0.74 ± 0.00 | 0.74 ± 0.00 | 2.03 ± 0.08 | 1.42 ± 0.03 | **0.30 ± 0.44** | 0.17 ± 0.33 |
| GBC | 0.74 ± 0.00 | 0.73 ± 0.00 | 51.26 ± 0.40 | **3.57 ± 0.30** | 6.32 ± 4.05 | 0.04 ± 0.04 |
| **Adult** | | | | | | |
| Mean CC compression rate: 0.33 ± 0.02 Mean CC compression time: 0.93 ± 0.05 | | | | | | |
| Linear SVC | 0.69 ± 0.19 | 0.67 ± 0.20 | 1.79 ± 0.08 | 1.53 ± 0.08 | **0.30 ± 0.84** | 0.18 ± 0.52 |
| Gaussian NB | 0.81 ± 0.01 | 0.81 ± 0.01 | 0.01 ± 0.00 | 0.93 ± 0.05 | **0.01 ± 0.00** | 0.02 ± 0.02 |
| RF | 0.86 ± 0.01 | 0.85 ± 0.01 | 2.04 ± 0.01 | **1.60 ± 0.09** | 2.11 ± 1.18 | 0.69 ± 0.59 |
| SVC | 0.76 ± 0.00 | 0.76 ± 0.00 | 81.70 ± 0.56 | 10.52 ± 2.31 | **5.06 ± 7.17** | 0.16 ± 0.19 |
| Tree | 0.81 ± 0.00 | 0.81 ± 0.01 | 0.12 ± 0.00 | 0.97 ± 0.05 | **0.10 ± 0.08** | 0.72 ± 0.72 |
| Logit | 0.80 ± 0.00 | 0.80 ± 0.00 | 0.08 ± 0.01 | 0.96 ± 0.05 | **0.05 ± 0.07** | 0.42 ± 0.68 |
| GBC | 0.86 ± 0.00 | 0.86 ± 0.00 | 2.33 ± 0.01 | **1.80 ± 0.09** | 2.37 ± 1.22 | 0.67 ± 0.57 |

**Notes.**

Linear SVC, linear support vector machine; Gaussian NB, naïve Bayes with Gaussian kernel; RF, random forest 100 CART trees; SVC, support vector machine with radial basis function kernel; Tree, CART decision tree; Logit, logistic regression; GBC, gradient boosting classifier with 100 CART trees.

ADULT data is pruned at 33%. CC compression rate is rather stable with only small standard deviation, but PS compression rate is characterised with huge variance. In this regard, complexity curve pruning is preferable as a more stable pruning criterion.

In all cases when training a classifier on the unpruned data took more than 10 s, we observed huge speed-ups. With the exception of SVC on LED data set, complexity curve pruning performed better than progressive sampling in such cases. Unsurprisingly, real speed-ups were visible only for computationally intensive methods such as Support Vector Machines, Random Forest and Gradient Boosted Decision Trees. For simple methods such as Naïve Bayes, Decision Tree or Logistic Regression, fitting the model on the unpruned data is often faster than applying the pruning strategy.

These results present complexity curve pruning as a reasonable model-free alternative to progressive sampling. It is more stable and often less demanding computationally. It does not require additional convergence detection strategy, which is always an important consideration when applying progressive sampling in practice. What is more, complexity curve pruning can also be easily applied in the context of online learning, when the data is being collected on the fly. After appending a batch of new examples to the data set, Hellinger distance between the old data set and the extended one can be calculated. If the distance is smaller than the chosen threshold, the process of data collection can be stopped.

## CONCLUSIONS

In this article, we introduced a measure of data complexity targeted specifically at data sparsity. This distinguish it from other measures focusing mostly on the shape of optimal decision boundary in classification problems. The introduced measure has a form of graphical plot—complexity curve. We showed that it exhibits desirable properties through a series of experiments on both artificially constructed and real-world data sets. We proved that complexity curve capture non-trivial characteristics of the data sets and is useful for explaining the performance of high-variance classifiers. With the conditional complexity curve it was possible to perform a meaningful analysis even with heavily imbalanced data sets.

We then demonstrated how complexity curve can be used in practice for data pruning (reducing the size of training set) and that it is a feasible alternative to progressive sampling technique. This result is immediately applicable to all the situations when data overabundance starts to pose a problem. For instance, it is possible to perform a quick exploration study on a pruned data set before fitting computationally expensive models on the whole set. Pruning results may also provide a suggestion for choosing proper train-test split ratio or number of folds of cross-validation in the evaluation procedure.

We argue that new measures of data characteristics, such as complexity curves, are needed to move away from a relatively static view of classification task to a more dynamic one. It is worth to investigate how various algorithms are affected by certain data manipulations; for example, when new data become available or the underlying distribution shifts. This would facilitate the development of more adaptive and universal algorithms capable of working in a dynamically changing environment.

Experiments showed that in the presence of large number of redundant attributes not contributing to the classification task complexity curve does not correlate well with classifier performance. It correctly identifies dimensional sparseness of the data, but that is misleading since the actual decision boundary may still be very simple. Because of this, as the next step in our research we plan to apply similar probabilistic approach to measure information content of different attributes in a data set and use that knowledge for performing feature selection. Graphs analogical to complexity curves and generalisation curves would be valuable tools for understanding characteristics of data sets and classification algorithms related to attribute structure.

Another limitation our method is the assumption of lack of attributes interdependencies. While the presence of small dependencies does not disrupt the analysis, when strong high

dimensional dependencies are present, the complexity curve does not correlate with classifier performance well. This means that it is infeasible to use for some domains; for example, highly epistatic problems in bioinformatics.

Our long-term goal is to gain a better understanding of the impact of data set structure, both in terms of contained examples and attributes, and use that knowledge to build heterogeneous classification ensembles. We hope that a better control over data sets used in experiments will allow to perform a more systematic study of classifier diversity and consensus methods.

### Funding

The study is cofounded by the European Union from resources of the European Social Fund. Project PO KL "Information technologies: Research and their interdisciplinary applications," Agreement UDA-POKL.04.01.01-00-051/10-00; Polish National Science Centre (grant numbers: 2015/16/T/ST6/00493 to JZ, 2014/15/B/ST6/05082 and 2013/09/B/NZ2/00121 to JZ and DP); EU COST BM1405 and BM1408 actions. The funders had no role in study design, data collection and analysis, decision to publish, or preparation of the manuscript.

### Grant Disclosures

The following grant information was disclosed by the authors:
European Union: UDA-POKL.04.01.01-00-051/10-00.
Polish National Science Centre: 2015/16/T/ST6/00493, 2014/15/B/ST6/05082, 2013/09/B/NZ2/00121.
EU COST: BM1405, BM1408.

### Competing Interests

The authors declare there are no competing interests.

### Author Contributions

- Julian Zubek conceived and designed the experiments, performed the experiments, analyzed the data, contributed reagents/materials/analysis tools, wrote the paper, prepared figures and/or tables, performed the computation work, reviewed drafts of the paper.
- Dariusz M. Plewczynski conceived and designed the experiments, analyzed the data, wrote the paper, reviewed drafts of the paper.

### Data Availability

Associated code is available to download at:
https://github.com/zubekj/complexity_curve.
(open source Python implementation).

## Supplemental Information

Supplemental information for this article can be found online at http://dx.doi.org/10.7717/peerj-cs.76#supplemental-information.

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
