# Peer review of "Complexity curve: a graphical measure of data complexity and classifier performance"

_PeerJ Computer Science, doi:10.7717/peerj-cs.76_

## Round 0.1 · original submission · Major Revisions

Please thoroughly address all the comments from the reviewers, especially in relation to experimentally comparing this work with previous work on data complexity (not just citing them) as well as using a much larger set of benchmark datasets.

The chosen UCI datasets have been already used for almost 20 years and are not representative of the current challenges faced in the machine learning field. Many more datasets, larger datasets (in both #instances and #attributes), and datasets facing some particular challenge (class imbalance, sparseness, multi-class, multi-label, etc) should be used.

Reviewer 1 ·

Basic reporting

This work introduces a new kind of plot to assess the complexity of a given dataset based on probability distribution and Hellinger distance. The resulting plot gives an idea about the complexity of the data w.r.t. the size of the input sample.

- In terms of English, this work contains multiple grammar mistakes. (e.g. Line 19: how it can (be) used, Line 22: treats, line 31: varies, etc.), and the writing needs to be improved, avoiding some awkward sentences.

- The literature covered in this work needs to be enlarged including related works in the application areas that the authors are then using in the experimental validation.

- The structure of the paper and the figures are ok.

Experimental design

Apart from the visualization on multiple UCI datasets, the authors present two main application of these data complexity curves: data pruning with complexity curves and generalization curves.

In general, I would say that the experimental design is appropriate.

Validity of the findings

To assess the validity of the findings of this new kind of plot, the authors need to include a comparison with other data complexity measures in the application domains considered.

Additional comments

Apart from including comparison methods, I think that the authors should include discussion (new section) on how this plot can help us when dealing for instance with imbalanced data, high dimensional data or unlabeled data (semi-supervised learning)?

Reviewer 2 ·

Basic reporting

Yes, the article is written clear English with commendably few typographical and grammatical errors (please see comments to authors below).

One of the main issues with this paper is related to the introduction. Although the aim is clearly stated, this is not precisely described by clear objectives. This has the effect of making the paper unnecessarily complex to read, which disrupts the flow of argument causing confusion in the methods presented. It would appear that there are five sequential objectives covering the following topics:
• Measurement of data saturation to determine how representative a sample is of the complete data
• Measurement of dataset complexity assuming linear separability
• Determine which technique is likely to suit a dataset for a given task, e.g. classification, regression and so forth.
• Determine how this measurement can be used for data reduction/halting of online training
• Test the performance of classification techniques in incomplete datasets by controlling the variance of data through the measurement technique

The introduction has two further issues: firstly, when discussing evaluating classifier perform much focus is on ROC curves, where not all classification approaches have a threshold parameter/value in order to generate such a curve. Secondly, the linearly separable assumption means that it is not an unbiased comparison of various classification algorithms. Comparing an algorithm that makes the linearly separable assumption with one that does not using a measurement that makes the same assumption is biased. In general, the topic of epistasis in datasets should be much more thoroughly addressed. Both toy problem domains, e.g. multiplexer problems, and real world problem domains e.g. bioinformatics, are highly epistatic. Selecting six ad hoc datasets without testing their epistatic contents is insufficient to draw conclusions on the effect of the assumption as the datasets may be linearly separable themselves. The use of toy datasets where epistasis can be measured is recommended.

The background section is not identified as such. There is much recent work on identifying the complexity of instance bases and the understanding of test domains that is missing, e.g. Smith-Miles, K. A., Baatar, D., Wreford, B. and Lewis, R., “Towards Objective Measures of Algorithm Performance across Instance Space”, Computers and Operations Research, vol. 45, pp. 12-24, 2014. It is noted that the literature review/references cease in 2013.

No, the paper does not confirm to the template below, but the problems with the flow of ideas would not be assisted by this broad outline. Numbering of sections, subsections and equations would certainly help the readability of the paper.
Author Cover Page (see above)
Abstract
No more than approx. 500 words (or 3,000 characters).
Headings in structured abstracts should be bold and followed by a period. Each heading should begin a new paragraph. For example:
Background. The background section text goes here. Next line for new section.
Methods. The methods section text goes here.
Results. The results section text goes here.
Discussion. The discussion section text goes here.
Introduction
Materials & Methods
Results
Discussion
Conclusions
Acknowledgements

Unfortunately, this paper was printed in black-and-white, which makes many of the figures time-consuming to interpret, e.g. figure 4. The style adopted in figure 7 and 8 is understandable. However, figure 11 is cluttered where more space can be given to the individual performance graphs. A figure illustrating the Hellinger distance is recommended as it is central to this work.

This paper does represent a coherent piece of work, which has been thoroughly investigated.

It appears that all raw data has been made available, although at this stage of revision the available code has not been verified.

In general, clear definitions and proofs are provided. Inconsistencies are noted in the comments to authors.

Experimental design

The most encouraging part of the paper is the primary research. It is still an open research question of value as to the best method to measure the difficulty of a dataset without applying the actual task to it, e.g. application of multiple classification algorithms to determine the difficulty of a dataset. The hypothesis presented in this work appears sound, although the supporting results could be improved in this paper.

Although the aim is well prescribed, the research objectives and corresponding research questions are not well defined. The complexity curve, conditional complexity curve and generalisation curve be clearly separated in terms of their objectives, otherwise they tend to confuse each other in purpose.

The investigation is rigourous, albeit missing the analysis of the effect of epistasis, which is considered vital to support these measures in real-world domains. The justification analysis of the selected UCI datasets should include what they contain as well as what they do not contain. The accuracy score on the led dataset across the different classification algorithms appears too consistent - it is worth noting that due to the inherent noise in the dataset that 74% is expected and ten fold cross validation means that there are no deviations greater than two decimal places occurring. Curiously, in large datasets a 70:30 split is more usual as much faster to process.

There are no ethical issues in this paper

Validity of the findings

More analysis of the data in terms of redundancy, irrelevancy and epistasis should be conducted. It is often considered that the BCW has irrelevant information, which the results indicate (page 24). It is recommended to cite past work where classification algorithms have identified the considered irrelevant data.

The datasets used are readily available as well as program code (not yet tested).

As there are no clear objectives and complementary research questions, the conclusions are not as clear as they could have been. The contributions to the field need to be more clearly identified and supported.

Any speculation is identified as such.

Additional comments

Major revisions are recommended due to the lack of investigation on the linear separability assumption in regard to epistasis. However, the underlying aim and hypothesis are worthwhile, which will lead to a strong publication.

The background could be strengthened on recent work on determining problem difficulty, e.g Smith-Miles, K. A., and Lopes, L. B., “Measuring Instance Difficulty for Combinatorial Optimization Problems”, Computers and Operations Research, vol. 39, no. 5, pp. 875-889, 2012. Comment on the importance of decision boundaries would be welcome. It is not clear what makes a previous method domain-specific rather than universal.

Please define all matrices, e.g. expected value E

How does the system address noise on the target variable instead of simply on the source variables? Please comment on whether the type of noise have any significant effect?

Please state what figure 1 illustrates rather than what it is and explain why it is not unimodal.

An introduction to the properties section is needed, albeit briefly.

Page twelve: please be consistent or state the difference between ‘q’ and ‘a’ in the probability equations

There appears to be a missed opportunity for additional insight in this paper. The UCI datasets are well studied in classification algorithms, where the past best performance can be cited, e.g. in a column in table 1. This can be compared with the output of the measurement curves to observe any consistencies/disagreements.

Is the speculation that there are no redundant variables in the wine or glass datasets supported by external evidence?

How does the introduced technique perform with datasets containing many outliers?

Minor comments
abstract: ‘our performance measure treats...’
Page: six: ‘training set’
seven: ‘is’ instead of ‘is is’
eight: ‘assess complexity’
ten: ‘for the comparison of algorithms’
sixteen: ‘usefulness of the chosen’
twenty-four: ‘relying heavily on’

·

Basic reporting

The structure of the paper is correct, although wording and clarity in the experimental results should be improved.

The authors present a method for assessing data complexity based on probability distributions and Hellinger distance and use it both as a classifier's performance measure and data pruning.

They claim to introduce a universal measure of complexity. What do they mean by universal? A single complexity measure can neither characterize the problem’s difficulty nor fully capture the relationship between the problem's difficulty and the learner’s ability.

The classifiers' accuracy depends both on the constraints of the algorithm and the intrinsic complexities of the data. Ho and Basu (see reference below) designed a set of complexity measures to evaluate the difficulty of the class boundary for classification problems. Ho’s study focused on the analysis of data complexity in order to find the relationship between the intrinsic geometries of data and the learners’ generalization mechanisms. The ultimate goal was to identify learners’ domains of competence so that one could select the most suitable learner for a given problem.

Ho, T. K. and Basu, M. (2002). Complexity measures of supervised classification problems. IEEE Transactions on Pattern Analysis and Machine Intelligence, 24 (3):289–300

I would suggest that the authors thoroughly review the literature on data complexity. Sotoca et al. (2005) is only one review that mainly introduces Ho’s complexity measures.

Experimental design

The authors claim that their measure can be applied not only to classification, regression, and clustering tasks but also to online learning tasks. However, they only provide empirical results on classification problems.
The authors performed experiments on a test bed composed of six data sets from the UCI Machine Learning Repository, which has 255 classification problems. How did the authors select these six data sets?
In addition, they cherry-picked data sets with no missing values, which is another aspect of data complexity. How the proposed measure can be a universal complexity measure if it is not able to cope with this type of data sets?

The authors consider WINE and GLASS data sets complex based on their ratio between the number of attributes and the number of instances. However, this ratio is, again, one dimension of complexity, which is not enough to fully describe the problem’s difficulty.

Sample size is another dimension of complexity. Actually, it is an extrinsic characteristic of a data set and it does not determine the problem complexity as stated in line 133.

Categorical variables should not be treated as continuous attributes, especially when distance metrics are involved.

Regarding the assumption that all attributes are independent, the authors indicate that their assumption “yields good results in practice”. However, the number of attributes of the data sets used in the experimentation ranges from 4 to 21, which is not large enough to demonstrate they can make such an assumption. What would be the impact on bioinformatic data sets usually described by hundreds of attributes?

I agree that there is a threshold where the more data the worse the learner’s performance is. The authors make an interesting point on how the evaluation of machine learning techniques is static both in parametrization and data.

Validity of the findings

Figure 11 shows the complexity curves obtained for different data sets and the performance of a set of learners run on these data sets. We observe that there is no correlation between the complexity measure and the performance obtained by the classifiers. A Hellinger distance does not discriminate between leaners’ performance. For a Hellinger distance of 0.8, the KNN reaches 90% accuracy on the Breast Cancer Wisconsin data set while it poorly performs (52% accuracy) on the Monks-1 data set.

On the other hand, are the learners tuned to work at their best performance for each data set? If the learners are run at their baseline configuration, the resulting comparison would not be fair.

A decision tree algorithm is run on the stripe and chessboard data sets. Which algorithm in particular? C4.5?
Why did the authors choose this combination: stripe/chessboard and decision tree?

Additional comments

I encourage the authors to keep researching on data complexity analysis which is crucial to gain understanding on learners’ behaviors. I would suggest that the authors narrow the scope of their study.

---

## Round 0.2 · Minor Revisions

The revision has been re-reviewed by Reviewer's 1 and 2. Reviewer two makes very good points about the wording of some parts of the manuscript to make it less misleading. Please, address these.

Reviewer 1 ·

Basic reporting

The authors have successfully addressed most of my previous comments. IN my opinion this new version of the manuscript is ready to be published.

Experimental design

It has been widely improved.

Validity of the findings

No extra comments.

Additional comments

No extra comments.

Reviewer 2 ·

Basic reporting

Please see previous comments. As noted by other reviewers, although the English is understandable it could be improved.

Experimental design

Please see previous comments that have been addressed

Validity of the findings

There is a query on target attribute noise and epistasis, please see below. This is so that readers can understand where the technique is appropriate and where caution is needed in its application.

Additional comments

This paper has been much revised with significant effort by the authors to address the reviewers’ comments. The addition of substantially more datasets, removal of ambiguity in equations and the addition of many insightful comments are commended. However, there are two areas of weakness that need addressing prior to publication.

1. The discussion of noise on the target variable is still not as clear as it could be presented.
Adding “...classes ambiguity (e.g. caused by noise on the target variable). " would help

*** How does the system address noise on the target variable instead of simply on the source variables? Please comment on whether the type of noise have any significant effect? ***
No, our measure disregards this type of noise. It is only concerned with sample sparsity. To make it clear, we added the following to the introduction: " In this article we introduce a new measure of data complexity targeted at sample sparsity, which is mostly associated with variance error component. We aim to measure information saturation of a data set without making any assumptions on the form of relation between dependent variable and the rest of variables, so explicitly disregarding shape of decision boundary and classes ambiguity. "

> Furthermore, there is an implication that noise comes from an overlap between classes, which is an odd implication considering noise as a signal that carries no useful information - some classes naturally overlap and it is useful to understand this information.
“… some irreducible noise. … partly from an overlap between classes.”

2. Epistasis and attribute dependence are not the same concept (although related).
*** In general, the topic of epistasis in datasets should be much more thoroughly addressed. Both toy problem domains, e.g. multiplexer problems, and real world problem domains e.g. bioinformatics, are highly epistatic. ***
We performed an experiment with an artificial data set "chessboard" with strong attribute dependencies (see section section "Complexity curve and the performance of an unbiased model"). In the "Generalisation curves for benchmark data sets" we added references supporting our claim of attribute interdependencies in monks-1 and car data sets. Moreover, we added two new sections ("Comparison with other complexity measures" and "Large p, small n problems") in which we present experimental results on a much larger collection of data sets, among them microarray data important for bioinformatics.
“In the introduced complexity measure we assumed independence of all attributes … Of course, if the independence assumption is broken bias error component may still be substantial.”

> The topic of epistasis is still not properly addressed as attribute dependencies are not the same as epistatic relationships although there is a connection. There are attribute dependencies in both the multiplexer and parity problem domains, but only the former is epistatic. In the latter, all attributes are equally dependent (important). In the former, the importance of an attribute varies with the value of another attribute and they are not all equally important. Frequentist approaches can successfully address problems with attribute dependencies that all equal in importance, but often struggle in epistatic domains, i.e. counting the occurrence of values in individual attributes without regard to epistatic relationships does not function well.
Please test your system on a range of multiplexer problems or explicitly state that the attribute independence assumption makes the approach unlikely to be suitable for highly epistatic problems.

One final thought, which is optional to address: the generalisation curve is claimed to be more insightful than the standard training progression curve, but they were never placed side-by-side for a direct comparison of usefulness. I am curious to know what the set of standard training progression curves would look like for figure 12 and how the new analysis/visualisation method specifically aids understanding/insight.

---

## Round 0.3 · Minor Revisions

I can see from the manuscript changes that the authors have cut out a significant chunk of the manuscript, related to the generalisation curves. Hence, some of the initial claims about the contribution of this work are not valid anymore. I think that the authors need to make a case for which their paper still deserves publication.

Moreover, I don't understand the line "Values larger than 0.22 or smaller than -0.22 are significant at a = 0:05 significance level." in the caption of tables 2 and 3. It does not match the rows marked as bold in the tables. Moreover, were corrections for multiple comparisons applied?

---

## Round 0.4 · accepted · Accept

I am happy with the latest changes that the authors have made to the manuscript.